# OpenReview forum: "Soft Instruction De-escalation Defense"
_ICLR.cc/2026/Conference — ICLR 2026 Conference Desk Rejected Submission_

### Official Review · Reviewer_F1H1 · 2025-10-28

**Soundness:** 3
**Presentation:** 3
**Contribution:** 3
**Rating:** 6
**Confidence:** 4

**Summary:**

The paper proposes a lightweight defense mechanism for LLM-based Agents, designed to resist Prompt Injection attacks.
The authors introduce an iterative and lightweight purification approach, which gradually rewrites potential instructions to neutralize malicious behavior.
This design avoids two common weaknesses in previous methods:

1. vulnerability to adversarial rephrasing, and
2. overly aggressive detection that causes frequent false positives.

Through evaluations on multiple LLMs, the authors demonstrate that SIC can reduce the Attack Success Rate (ASR) to zero across all tests while preserving the Agent’s original task performance—showing the clear effectiveness of the method.
Additionally, the work emphasizes parallelization, and the reduced computational cost makes it well-suited for real-world deployment.

**Strengths:**

1.	By employing a preprocessing module or introducing an LLM-as-a-Judge component, the method avoids modifying internal model parameters, making it friendly to black-box models.
2.	The multi-round strategy is more robust than a one-shot approach, and experimental results strongly demonstrate the effectiveness of this iterative mechanism.
3.	The method is efficient and parallelizable, while maintaining the original task performance, which is crucial for its practical applicability.

**Weaknesses:**

1.	Some detector or rewriter designs rely on external LLMs. Has the paper considered the scenario where the attack itself targets these external LLMs? Could this lead to delayed defense response or even worse cascading failures?
2.	The performance of both the rewriter and the detector depends heavily on the quality of their prompt templates. Combined with the first concern, how does the framework ensure the robustness and diversity of these templates under adversarial conditions?
3.	Regarding the choice of the auxiliary LLM, is there any ablation study or selection criterion provided? How do the alignment quality and model size of the auxiliary LLM affect the performance and latency of the SIC framework?

**Questions:**

Please see weaknesses.

---

> ### Author Response · Authors · 2025-11-20
>
> Dear reviewer,
>
> thank you for your thoughtful comments and positive assessment. We are glad you found the approach, robustness, and efficiency promising. Below, we address your points in detail.
>
> 1. Yes, if the LLM provider itself is attacked or degraded, our defense can suffer delayed responses. However, in our setup, we use the same LLM for both the agent and the defense, so the whole system is affected in a coupled manner. In the general case, this primarily leads to DoS-like behavior (slower or more frequently halted interactions), while the defense logic itself remains robust rather than being easily bypassed.
> 2. We agree that the performance of both the rewriter and the detector depends on the prompt templates. We experimented with a range of prompts – from very short, almost one-line instructions to more elaborate descriptions – and found that a middle ground, which is what we use (see Appendix), works best. For example, if we remove the sentence “You must make sure that the instructions are really instructions, not just information.” from the detector prompt, utility decreases because the system produces more false positives on information that only looks like instructions (e.g., “Hey, listen, did you hear that X has happened”), even though such patterns can also become attack vectors. This illustrates the inherent trade-off between security and utility. At the same time, this sensitivity is also an advantage: prompt templates can be patched and iterated on at very low cost, allowing practitioners to tune SIC for their own domains and threat models.
> 3. Regarding the auxiliary model, in all our experiments, we simply use the same model that is being evaluated; in that sense, SIC is a “self-defense” setup. The rationale is that what ultimately matters is whether that particular model would execute the attack, so we want the defense to reflect the model’s own notion of “instruction-like” content. We will make this explicit in the paper. A more systematic ablation over different auxiliary models (varying size and alignment quality) is an interesting direction for future work; our expectation is that stronger, better-aligned auxiliary models would further improve robustness at the cost of higher latency.

---

### Official Review · Reviewer_Eo7B · 2025-10-31

**Soundness:** 3
**Presentation:** 3
**Contribution:** 3
**Rating:** 6
**Confidence:** 3

**Summary:**

This paper primarily investigates the performance of different defense mechanisms against various instruction injection attacks, with a focus on analyzing the robustness of the SIC method across multiple models and tasks. The article demonstrates that SIC can reduce attack success rates to 0% under various attack types while causing only minimal degradation in utility. Through ablation studies, it explores the impact of rewriting and chunking mechanisms, explaining how the chunking mechanism helps reduce attack success rates while also noting the potential false positive issues it may introduce.

**Strengths:**

1. The SIC method maintains 0% ASR across various attack types and models, demonstrating significant robustness.

2. The comparative analysis system is comprehensive, including different models and existing defense methods.

3. Ablation experiments are included, explaining the underlying reasons why the chunking mechanism reduces attack success rates.

**Weaknesses:**

1. The analysis of false positive sources is relatively brief, only mentioning "instruction-like statements," lacking more specific classification or mitigation strategies.

2. The experiments primarily focus on plaintext prompt injection, lacking validation against more covert multi-turn or cross-modal attacks.

3. There is insufficient detailed evaluation of defense overhead, such as computational resource consumption and response latency.

4. In multilingual environments, can SIC still effectively identify and intercept instruction injections?

**Questions:**

see Weakness Section

---

> ### Author Response · Authors · 2025-11-20
>
> Dear reviewer,
>
> thank you for your thoughtful comments and positive assessment. We are glad you found the robustness results, comparisons, and ablations useful. Below, we address your points in detail.
>
> 1. The primary source of false positives we observed originates from benign, notification-style sentences that resemble instructions to the classifier/rewriter. For example: “Please note that your rent has been increased.” In our experiments, this was essentially the only systematic failure mode of the combination of the classifier and the rewrite pipeline.
> 2. You are right that our experiments focus on single-turn, plaintext prompt injection. We chose this setting because, even in this basic case, it remains unsatisfactorily solved, which is exactly what AgentDojo (the popular benchmark we evaluate on) targets. Multi-turn and cross-modal attacks are very interesting directions and we are actively exploring them, but it is not yet clear what the “right” notion of rewriting is for, e.g., images or mixed-modality content.
> 3. We did not include a detailed latency comparison for two reasons. First, most of our experiments were conducted via OpenRouter, where we observed substantial variance in latency across runs and even within different times of day, which makes fair wall-clock comparisons between methods quite noisy. Second, many of the other defenses we compare against use specialized, smaller models or simpler LLM calls, so they are indeed faster in the simplest threat setting. Thus, for basic attacks, the efficiency–security trade-off can lean toward these lighter defenses. However, for stronger adversaries (see the new results in Section 6 in the updated manuscript), our method is clearly more robust, making the overall security–utility–efficiency trade-off more favorable for SIC in higher-threat regimes. We will clarify this and, where space permits, add more concrete discussion on latency.
> 4. We conducted some preliminary tests where the attack instructions were written in different languages, and we did not observe a higher ASR compared to English. Based on this, we would expect the performance of SIC to be similar in a truly multilingual setup, provided the underlying model has comparable capabilities across languages. However, to the best of our knowledge, there is currently no established multilingual benchmark for agentic prompt injection; therefore, we did not conduct a systematic evaluation.

---

### Official Review · Reviewer_m4mo · 2025-10-31

**Soundness:** 3
**Presentation:** 3
**Contribution:** 3
**Rating:** 6
**Confidence:** 3

**Summary:**

This paper proposes Soft Instruction Control (SIC), a defense method designed to counter indirect prompt injection attacks in LLM agents. SIC performs iterative instruction detection and rewriting to “soften” and sanitize input data, effectively preventing malicious instructions from being activated during the agent’s execution phase. Experiments conducted on the AgentDojo benchmark, covering various models and attack scenarios, demonstrate that SIC can significantly reduce the attack success rate (ASR) to 0% while maintaining high task utility.

**Strengths:**

- SIC combines iterative rewriting and detection to establish a “soft control” defense mechanism. This design balances defense effectiveness with performance and offers strong deployment feasibility.
- Extensive evaluations were conducted on the AgentDojo benchmark, covering various models and attack scenarios. Comparisons with other defenses, such as MELON and PI-GUARD, further validate the effectiveness of SIC.

**Weaknesses:**

- The paper presents a theoretical analysis of SIC’s latency but lacks experimental comparisons with other defense methods. This limitation makes it difficult for readers to fully assess SIC’s overall performance across the “security–utility–efficiency” trade-off. It is recommended to include detailed latency comparison experiments among different defense methods to further strengthen the practical validation of the approach.
- The paper does not specify the exact auxiliary LLM model used in the SIC method. Since the performance of SIC may depend on the capability of the auxiliary model, it is recommended to include ablation studies to quantitatively illustrate the direct impact of the auxiliary model’s performance on SIC’s defense effectiveness.

**Questions:**

- Do the results of the adaptive attack reported in Section 6 (ASR = 60% in the Slack scenario) generalize to other tasks and models? Could more experimental results be provided?
- What is the defense cost associated with the auxiliary model used in SIC? Would training or fine-tuning a lightweight model separately reduce the defense cost or enhance the defense effectiveness?

---

> ### Author Response · Authors · 2025-11-20
>
> Dear reviewer,
>
>
> Thank you for your thoughtful comments and positive assessment. We are happy that you found the contribution, presentation, and evaluation convincing. Below, we address your points in detail.
>
> 1. We did not include a direct latency comparison for two reasons. First, most of our experiments were run via OpenRouter, where we observed substantial variance in latency across runs and even across times of day, which made fair comparisons between methods quite noisy. Second, the other defenses we compare to typically use specialized, smaller models or simpler LLM calls, so they are indeed faster in the simplest setting. Thus, for basic attack scenarios, the efficiency–security trade-off can lean toward these lighter defenses. However, for stronger adversaries (see the new results in Section 6), our method is clearly more robust, making the overall security–utility–efficiency trade-off significantly more favorable for SIC in the higher-threat regime. We will clarify this and, as far as possible, add more concrete latency discussion.
> 2. Regarding the auxiliary model, in all our experiments, we simply use the same model that is being evaluated; in that sense, SIC is a “self-defense” setup. The rationale is that what ultimately matters is whether a given model will execute the attack. We will make this explicit.
>
> **Questions:**
>
> Yes, we provided extra results, although on the GPT-5-mini model, where we see that SIC performs significantly better ($3\times$) than all the other methods. This also demonstrates that the defense strength depends both on the auxiliary model and the target model, as expected.
> The idea of using a separate lightweight auxiliary model is very interesting. In principle, one could train or fine-tune a smaller model, or even use an ensemble of smaller classifiers, for the detection component to reduce cost, while keeping the rewriting behavior as close as possible to the main model. The rewriting part is harder to distill faithfully, but with continuing improvements in inference speed and efficiency, we expect the overall latency of SIC to improve.

---

### Official Review · Reviewer_CRvw · 2025-11-01

**Soundness:** 2
**Presentation:** 3
**Contribution:** 3
**Rating:** 6
**Confidence:** 3

**Summary:**

This paper proposes Soft Instruction Control (SIC), an iterative sanitization defense against prompt injection attacks in tool-augmented LLM agents. The core idea is to repeatedly rewrite untrusted input to remove imperative instructions, inject canary instructions to detect rewriting failures, and use chunked classification to verify no malicious instructions remain. The method is evaluated on AgentDojo benchmark across multiple models (GPT-4o, Qwen3-32B, Kimi-k2, GPT-4.1-mini) and achieves 0% ASR on standard attacks while maintaining reasonable utility. However, under adaptive attacks (Nasr et al., 2025), the defense achieves only 60% ASR, revealing three primary failure modes: embedded executable workflows, authority-styled language, and partial-failure narratives.

**Strengths:**

1. Combining rewriting, canary detection, and chunked classification creates defense-in-depth that is harder to bypass than single-layer approaches.
2. Testing across multiple SOTA models (GPT-4o, Qwen3-32B, Kimi-k2, GPT-4.1-mini) and three task domains demonstrates generalizability on standard benchmarks
3. Unlike many security papers, the authors conduct worst-case analysis and clearly document three failure modes with concrete examples
Strong performance on standard attacks: Achieving 0% ASR on AgentDojo attacks while maintaining 50-55% utility is impressive compared to baselines

**Weaknesses:**

1. The paper assumes white-box access (Section 3) but the adaptive attack reveals the defense relies on assumptions (e.g., "instructions are imperative") that adversaries can trivially violate. The threat model should explicitly state what adversarial capabilities are not covered.
2. Section 4.2 claims "latency remains small" but provides no actual measurements. For production systems processing thousands of requests, the cost of R+1+k LLM calls per input could be prohibitive.

**Questions:**

1. Why is there no comparison with CaMeL (Debenedetti et al., 2025)? This is the primary related work and claimed inspiration. What are the trade-offs between SIC's soft approach and CaMeL's formal decomposition in terms of both security and utility?
2. What happens if an attacker discovers the specific canary text? Have you tested with randomized canaries or multiple diverse canaries? How does performance change?

---

> ### Author Response · Authors · 2025-11-20
>
> Dear reviewer CRvw,
>
> we thank you for your thoughtful comments. We are happy that you found the contribution and presentation to be good and appreciated our worst-case analysis and failure-mode study. Below, we address them in detail.
>
>
> **Weaknesses:**
>
> We, like other defenses [1,2,3], implicitly assume that there will be instruction-like components that cause the agent to execute a malicious action. We will make this assumption more explicit in the attack model.
> Regarding cost and latency, we agree that many rewrites can be expensive. In our setting, however, a single rewrite is usually sufficient and, for the average banking prompt, corresponds to roughly 320 generated tokens, i.e., about 2 seconds at 160 tokens per second. With more modern accelerators that reach up to 1000 tokens per second, the same rewrite would take only about 0.32 seconds—around 6× faster—so even 2–3 rewrites would add only ≈0.64–0.96 seconds of latency. In an optimized production environment, this added latency is small, especially given the strong performance improvements over other defenses (see new results in Section 6). Next, the rewriting task could be handled by a small specialized LM, which has the benefit of both efficiency and robustness to prompt injection (since it is not following arbitrary instructions by default). We leave this open to future work.
>
> **Questions:**
>
> 1. Indeed, CaMeL serves as a motivation, but it is a very different class of defense.
> Our method and the baselines we compare against operate only on raw LLM inputs/outputs and follow the paradigm of detecting and neutralizing prompt injections at the text level. In contrast, CaMeL makes a strict assumption that control flow is independent of data flow, which we deliberately avoid.
> Moreover, much of CaMeL’s strength comes from hand-engineered security policies and a full system built around the environment, including explicit data and control-flow tracking. This brings a significantly higher setup and integration overhead, whereas our method fits directly into a typical inference pipeline as a lightweight, plug-and-play component.
>
>     Nonetheless, we ran initial experiments using CaMeL on the banking dataset with the ImportantInstructions attack on GPT-4o. CaMeL achieves a clean utility of 56.25% (SIC 80%), a utility under attack of 55.55% (SIC 55.55%), and an ASR of 2.77% (SIC 0%). We also observed that CaMeL was surprisingly expensive i.e. ~30\$ in credits for this setting, while SIC cost around ~20\$.
>
> 2. In terms of security, knowing the canary could in principle have an effect. Empirically, in our adaptive attack setup the attacker does know the canary, yet still fails to achieve a high ASR in the new results (see Section 6 in the updated manuscript). We also experimented with varying the canary text and did not observe a difference in performance, which is why we chose the simplest fixed canary in the main experiments. For adversaries specifically targeting the rewriter, using multiple and randomized canaries may be an effective hardening measure with almost no additional cost.

---

### Note · Program_Chairs · 2026-01-17
**Submission Desk Rejected by Program Chairs**

The following references in this submission do not refer to real documents and/or have major errors in bibliographic information:

 Luca Debenedetti et al. Llm agents in the wild: Security implications of open-ended tool use. arXiv
preprint arXiv:2401.02345, 2024b.